# Fire-Induced Alterations of Soil Properties in Albic Podzols Developed under Pine Forests (Middle Taiga, Krasnoyarsky Kray)

**Alexey A. Dymov** [1,2,*], **Viktor V. Startsev** [1], **Evgenia V. Yakovleva** [1], **Yurii A. Dubrovskiy** [1], **Evgenii Yu. Milanovsky** [3], **Dariy A. Severgina** [1], **Alexey V. Panov** [4] and **Anatoly S. Prokushkin** [4]

[1] Institute of Biology Komi Scientific Center Ural Branch of Russian Academy of Science, Kommunisticheskaya 28, 167982 Syktyvkar, Russia

[2] Department of Physics and Soil Reclamation, Faculty of Soil Science, Lomonosov Moscow State University, 119991 Moscow, Russia

[3] Institute of Physicochemical and Biological Problems in Soil Science, RAS, Institutskaya Str., 2/2, 142290 Pushchino, Russia

[4] V. N. Sukachev Institute of Forest SB RAS, Akademgorodok 50/28, 660036 Krasnoyarsk, Russia

[*] Correspondence: dymov@ib.komisc.ru or aadymov@gmail.com; Tel.: +7-495-939-42-07 or +7-8212-24-51-15

**Abstract:** Fires are one of the most widespread factors of changes in the ecosystems of boreal forests. The paper presents the results of a study of the morphological and physicochemical properties and soil organic matter (SOM) of Albic Podzols under pine forests (*Pinus sylvestris* L.) of the middle taiga zone of Siberia (Krasnoyrsky kray) with various time passed after a surface fire (from 1 to 121 years ago). The influence of forest fires in the early years on the chemical properties of Albic Podzols includes a decrease in acidity, a decrease in the content of water-soluble compounds of carbon and nitrogen and an increase in the content of light polycyclic aromatic hydrocarbons (PAHs) in organic and upper mineral horizons. Podzols of pine forests that were affected by fires more than forty-five years ago are close to manure forest soils according to most physical and chemical properties. Significant correlations were found between the thickness (r = 0.75, $p < 0.05$), the moisture content (r = 0.90, $p < 0.05$) of organic horizons and the content of ∑PAHs in the organic horizon (r = −0.71, $p < 0.05$) with the time elapsed after the fire (i.e., from 1 to 121 years). The index of the age of pyrogenic activity (IPA) calculated as the ratio of ∑ PAHs content in the organic horizon to ∑ PAHs at the upper mineral horizon is significantly higher in forests affected by fires from 1 to 23 years than for plots with «older» fires (45–121 years). Thus, the article presents the conserved and most changing factors under the impact of fires in the boreal forests of Russia.

**Keywords:** boreal forest; wildfire; C; N; stable isotopes; PAHs

## 1. Introduction

Fires are one of the most important disturbance factors affecting Pine forests and development of Podzols during Holocene [1–3]. Fires in boreal forests, particularly in pine forests, are a natural historical factor in their development [4–6]. The pine lichen forests growing on soils with sandy and sandy loam textures are most affected by fires [7–10]. The light texture of the bed rocks leads to well-drained soils and organic horizons drying out, and provides the highest flammability [11]. Repeated forest fires are a significant factor for the co-existence of both Pine forests and Albic Podzols [12,13]. According to Soil Taxonomy [14], this soil is classified as Spodosols which refers to the red, brown or black (Bs) horizon below the light colored near-surface layer. Under acidic conditions, aluminum, iron and organic compounds migrate from the bleached eluvial horizon (E) down to the Bs horizon with percolating rainwater [15]. The spodic horizon (Bs) has high concentration amorphous mixtures of organic matter and aluminum, with or without iron, which have accumulated. Podzols are found only in coarse-textured and base-poor parent materials—most often sands and sandy tills [16]. Significant areas of Podzols are

represented in the boreal forest regions. Podzols cover about 485 million ha worldwide, mainly in the temperate and boreal regions of the Northern Hemisphere [17]. In the Krasnoyrsky kray, Podzols cover 3.7% (86 tsh. km$^2$) of territory [18].

Fires can affect morphological and physicochemical properties of soil [10,11,19], that persist for several thousands of years [20]. In the soils of recently burnt areas, a decrease in acidity [7] and an increase in ash elements affecting the exchange properties of soils were noted [21]. Wildfires in the boreal forest can affect SOM, including its resistance to decomposition [22,23]. In the short-term, large quantities of SOM C are released into the atmosphere when the vegetation biomass and SOM on the soil surface are combusted.

The pyrogenically altered carbon compounds can be stable and play a significant role in the global carbon cycle [24–26]. Santin et al. [27] showed that soils contain a significant concentration of pyrogenic carbon (PyC). Some of the organic compounds formed during the cyclization of polymer molecules can be sufficiently resistant to decomposition [26,28,29]. Fires can change the carbon and nitrogen isotopic composition [30]. Yet, data on the long-term effect of pyrogenesis on the isotopic composition of carbon and nitrogen have not been found in the literature. Polycyclic aromatic hydrocarbons (PAHs) and benzene polycarboxylic acids (BPCAs) are often used as fire biomarkers in soil organic matter [31–35]. According to Rey-Salgueiro et al. [36], the greatest PAH concentrations are formed at temperatures up to 600 °C. PAHs are usually formed under biomass combustion with high lignin content [37]. PAHs are known to have cancerogenic, mutagenic and toxic properties [38] and are also often used for estimating pyrogenic impact on soil organic matter due to their aromatic structure [35]. The forests of Siberia are most susceptible to fires in the boreal zone of Eurasia, but assessments of changes in the properties of soils in this region are extremely rare [39].

The aim of the work is to assess changes in Albic Podzols of the middle taiga of Siberia under the influence of surface fires and during post-fire succession.

## 2. Materials and Methods

### 2.1. Area Description, Soil Sampling, Geobotanical Descriptions and Fire Event-Dating

The study area is located in the middle taiga subzone nearby the ZOTTO International Observatory (60°26′ N, 89°24′ E) (Figure 1). The mean annual air temperature in the Zotino region (Krasnoyrsky kray) according to the Bor weather station is −3.5 °C. The sum of temperatures above 10 °C varies from 1200 to 1400 °C. The annual rainfall is 594 mm.

In this study, in the summer period of 2019, we have selected five lichen pine forests affected by fires from 1 to 121 years ago. Photos of landscapes and soil profiles are presented on Figure S1. In each site, we have conducted vegetation descriptions (reveles) on 20 m × 20 m square plots using the standard geobotanical methods [40]. Undergrowth, herb-dwarf shrub and moss-lichen strata were characterized by the relative abundance of species and the total projective cover (TPC) of each. Latin names are given according to http://www.worldfloraonline.org/ (access date 25 December 2022).

Soil profiles were investigated in each square plot up to the 100–120 cm depth. Two more random soil profiles of up to 40 cm depths were excavated in the center of round inventory plots. Soil samples were taken from each genetic horizon. Organic layers were collected in triplicate from a 20 cm × 20 cm area. Mineral soil samples for chemical analyses were collected from five points from the pit and then the samples were mixed. The field diagnostics were performed according to the World Reference Base for Soil Resources [17]. The color of the horizons was determined according to the Munsell soil color chart [41]. We added «pyr» to the horizon name when, during the field description, inclusions associated with fires were diagnosed (coal, soot, etc.).

The tree-ring core samples from three to five cuts' cross-sections for estimation fire scars were obtained according to [42–44].

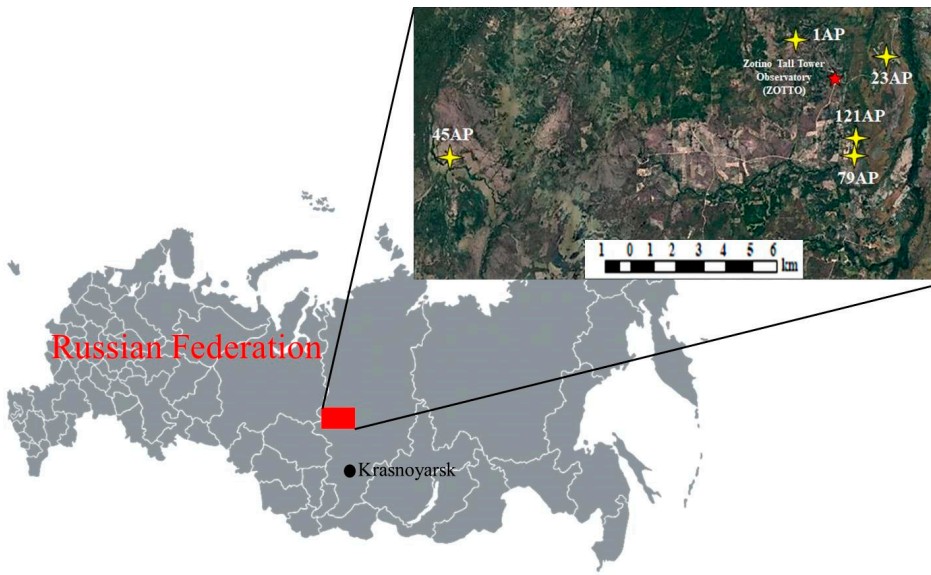

**Figure 1.** Location of study sites (the scheme is based on http://www.glovis.usgs.gov, accessed on 30 December 2022).

### 2.2. General Soil Analysis

Soil samples were air-dried under root temperature and sieved through a sieve with a mesh size of 2 mm. Pyrogenic and organic samples were crushed with a laboratory grinder. Chemical analyses of the soils were performed according to classical methods [45]. The methods of classical analyses are described in detail in our earlier works [10,35]. Specific surface area (SSA) was determined on a surface area analyzer (Sorbtometr-M, Katakon, Russia) with $N_2$ as the adsorbate gas at the Faculty of Soil Science, Lomonosov Moscow State University (Moscow, Russia). The method was described in detail previously [35].

### 2.3. Soil Organic Matter and PAHs Analysis

$C_{tot.}$ and $N_{tot.}$ were determined by dry combustion on an EA-1100 analyzer (Carlo Erba, Milano, Italy). Densiometric fractionation was performed with a sodium polytungstate (SPT0) solution with a density of $1.60 \pm 0.03$ g cm$^{-3}$ according to the method proposed [46] with recommendations [47].

The stable isotope ratios $^{13}C/^{12}C$ and $^{15}N/^{14}N$ were determined by an IsoPrime 100 isotope ratio mass-spectrometer (IsoPrime Corporation, Cheadle, UK) and vario ISOTOPE cube elemental analyzer (Elementar Analysen systeme GmbH, Hanau, Germany).

Water-soluble organic carbon (WSOC) and nitrogen (WSON) were analyzed according to the previously described method [12]. The extraction of PAHs from the soils was made on an Accelerated Solvent Extractor 350 (Thermo Scientific™, Waltham, MA, USA) in the Chromatography Collective Use Center of the IB FRC Komi SC UrB RAS. The method was described in detail previously [2].

### 2.4. Statistics

Significance differences and correlation coefficients were calculated using the STATISTICA 10.0 (Stat. Soft Inc., Tusla, OK, USA) software. Differences were considered significant at the significance level $p < 0.05$.

## 3. Results

### 3.1. Vegetation at the Study Sites

All study sites represent postfire development stages of Scots pine forests dominated in ground cover vegetation by ericoids *Vaccinium vitis-idaea*, *Ledum palustre* and lichens *Cladonia arbuscular*, *C. rangiferina* and *C. crispata*. The plant communities have similar species

composition and structure, and belong to the association Pinetum vacinioso-cladinosum (Table 1).

**Table 1.** Total projective cover (TCP) and dominant species of lower layers of plant communities of Pinetumvaccinioso-cladinosum formed on Albic Podzols.

| Age from Last Fire, Years | Total Projective Cover (TCP) and Dominant Species of Lower Vegetation Layers |
|---|---|
| 1 | Dwarf-shrub herb layer <br> 3%, *Vaccinium vitis-idaea*, *Calamagrostis* sp., *Chamaenerion angustifolium* <br> Moss-lichen layer <br> single burned residues of lichens |
| 23 | Dwarf-shrub herb layer <br> 5%, *Vaccinium vitis-idaea*, *Calamagrostis obtusata* <br> Moss-lichen layer <br> 50%, *Cladonia arbuscula*, *Cladonia rangiferina*, *Cladonia crispata*, *Cladonia cornuta* |
| 45 | Dwarf-shrub herb layer <br> 30%, Vaccinium vitis-idaea, Ledum palustre, *Vaccinium uliginosum* <br> Moss-lichen layer <br> 90%, *Cladonia arbuscula*, *Cladonia rangiferina*, *Cladonia crispata*, *Pleurozium schreberi* |
| 79 | Dwarf-shrub herb layer <br> 25%, *Vaccinium vitis-idaea. Calamagrostis obtusata, Diphasium complanatum, Vaccinium myrtillus* <br> Moss-lichen layer <br> 80%, *Cladonia arbuscula*, *Cladonia rangiferina*, *Cladonia crispata*, *Pleurozium schreberi* |
| 121 | Dwarf-shrub herb layer <br> 5%, *Vaccinium myrtillus*, *Vaccinium vitis-idaea*, *Ledum palustre*, *Diphasium complanatum*, *Empetrum hermaphroditum* <br> Moss-lichen layer <br> 90%, *Cladonia arbuscula*, *Cladonia rangiferina* |

Only single vascular plants occurred at the study plots at the initial stage of post-fire restoration (1 year after a fire, 1AP). Then, during the post-fire succession, the TPC of herb-dwarf shrub layer increases, reaching 20–30% at the plots burned 45–79 years ago (45AP and 79AP). However, the TCP of the herb-dwarf shrub layer on the 121AP plot decreases to only 5%.

However, the TCP of the herb-dwarf shrub layer on the 121AP plot decreases to only 5%. In the moss-lichen cover, there were no living mosses and lichens at the plot 1AP due to high intensive burning (fire scars on pine tree trunks are up to 10–12 m high). At the plot burned 23 years ago, total projective cover of the moss-lichen layer already reached 50% and further increased up to 80–90% at the plots burned out 45–121 years ago. The species diversity of vascular plants, mosses and lichens increased during the post-fire succession from 1–2 species at the 1AP to 8–9 species per plot at the later stages (Table 1).

### 3.2. Morphological Properties of Soils

The Albic Podzols surveyed in the studied fire chronosequence (from 1 to 121 year old) have the typical morphology for medium to high developed soils on well-drained coarse textured sands. Soil profiles have the following vertical structure: Oi/$Q_{pyr}$—Oe, $_{pyr}$—$E_{pyr}$—E—Bs—B. In mature pine forests (45–121AP), the soil organic horizon ($5.7 \pm 1.2$ to $9.7 \pm 0.6$ cm thickness) consists of several subhorizons composed of by plant debris with different stages of decomposition. The upper subhorizon Oi (1 to 3 cm deep) appeared already in the early stage of post fire succession (23AP) and consists of poorly decomposed remnants of ground vegetation and trees. The bottom Oe, $_{pyr}$ subhorizon in all studied plots contains plant residues at an average degree of decomposition with usually high values of charcoal particles of different sizes. A light gray-brownish (10YR4/1-10YR6/1) sandy horizon $E_{pyr}$ is formed below the organic horizon with a capacity of 4 to 6 cm and is enriched with dark colored organic matter and charcoal. The whitish, light gray (7.5YR8/1-10YR8/1) horizon E (6 to 29 cm thickness) that appeared below does not have visible structures and irregularly contain charcoal particles. The spodic horizon Bs (10 to

40 cm thickness) is yellowish brown (7.5YR6/8-10YR6/8), sandy, unstructured and is characterized by the accumulation of SOM forming complex compounds with Al and Fe. The deepest B horizon is non-uniform in color (from 2.5.6/6-2.5Y7/4 to 10YR 4/6) and is also without structures, but in the same case, this horizon can be more heavy-textured. Clay-like material underlaid the sandy B horizon at site 45AP.

The moisture content and thickness of the organic horizons are important characteristics of the fire's effect (Figure 2a). The thickness of the organic horizon $Q_{pyr}$ in the area affected by fire a year ago (1AP) was minimal due to complete or partial combustion of lichen and organic layers, and was presented by charred organic material. Along with post-fire recovery of ground vegetation and the pine canopy, the thickness of the organic horizons gradually increased with age because of the fire impact, due to the accumulation of decay of ground cover plants and trees. The largest values of the thickness of the entire organic horizon were measured for the soil of the oldest site 121AP (8.3 ± 2.9 cm).

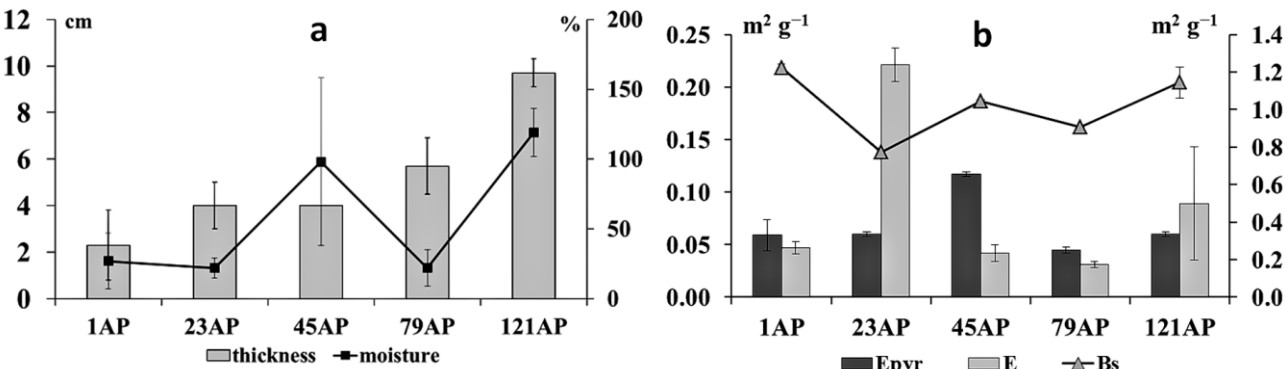

**Figure 2.** The thickness (left axis) and moisture content (right axis) of the organic horizons in the study soils (**a**). Specific surface area (**b**) of upper horizons' studied soils. E and $E_{pyr}$ on the left axis; organic layer and Bs on the right axis.

The lowest moisture content was found in the 1AP (WSC = 27%) and 23AP (22%) plots, 1 and 23 years after the fire, respectively (Figure 2a). Surprisingly low soil moisture content (22%) was also found at the 79 AP site. The highest WSC (119 ± 17%) was found in organic horizons of the oldest forest affected by a fire 121 years ago.

The correlation analysis for the $Q_{pyr}$ (Oe) pyrogenic horizon showed a significant positive relationship between the thickness (r = 0.75, *p* < 0.05) and the moisture content of the horizon (r = 0.90, *p* < 0.05) with the time elapsed after the fire (i.e., from 1 to 121 years).

### 3.3. The Main Chemical Soil Properties

The investigated Podzols have typical chemical properties (Table 2). The soil horizons vary from slightly ($pH_{H2O}$ 6.1) to strongly acidic ($pH_{H2O}$ 4.1). The lowest $pH_{H2O}$ was observed in organic horizons, where the $pH_{H2O}$ values varied from 4.1 to 5.6. The highest $pH_{H2O}$ (5.6) was observed in the Q layer (charred organic material) of the 1AP site. Surprisingly, the most acidic condition among analyzed soils was the organic layer from the 23AP site ($pH_{H2O}$ 4.1). Organic layer Oi and Oe subhorizons in the oldest 121AP site have a $pH_{H2O}$ of 4.4 and 4.2, respectively. Those are characterized as the lowest $pH_{KCl}$. Intermediate stages of post-fire restoration demonstrated a relatively narrow variation range of $pH_{H2O}$ values between 4.6 and 4.8.

**Table 2.** Chemical parameters of the studied soils.

| Site | Horizon | Depth, cm | pH H$_2$O | pH KCl | C$_{tot.}$ (g kg$^{-1}$) | N$_{tot.}$ (g kg$^{-1}$) | C/N | WSOC (mg g$^{-1}$) | WSON (mg g$^{-1}$) | Ca$^{2+}$ (mmol 100 g$^{-1}$) | Mg$^{2+}$ | K$^+$ | Na$^+$ | $\sum$ | BS, % | CEC |
|---|---|---|---|---|---|---|---|---|---|---|---|---|---|---|---|---|
| | Q$_{pyr}$ | 0–1 | 5.6 | 4.1 | 448 ± 16 | 19.4 ± 2.1 | 27 | 0.46 ± 0.09 | 0.04 ± 0.01 | 10.26 | 0.828 | 1.393 | 0.057 | 12.54 | 23 | 54.0 |
| | E$_{pyr}$ | 1–6 | 4.7 | 3.4 | 34 ± 5 | 0.93 ± 0.19 | 43 | 0.05 ± 0.01 | 0.005 ± 0.001 | 0.17 | 0.009 | 0.033 | 0.005 | 0.22 | 8 | 2.8 |
| | E | 6–17 | 5.8 | 4.7 | 2.7 ± 0.6 | 0.25 ± 0.07 | 13 | 0.03 ± 0.01 | 0.004 ± 0.001 | – | – | 0.001 | – | – | 1 | 0.2 |
| 1AP | Bs1 | 17–30 | 5.5 | 5.0 | 7.9 ± 1.8 | 0.50 ± 0.10 | 18 | 0.022 ± 0.004 | 0.003 ± 0.001 | – | – | 0.011 | – | 0.01 | 1 | 1.0 |
| | Bs2 | 30–50 | 5.8 | 4.9 | 3.0 ± 0.7 | 0.32 ± 0.09 | 11 | 0.020 ± 0.004 | 0.008 ± 0.002 | – | – | 0.016 | 0.004 | 0.04 | – | – |
| | B1 | 50–80 | 5.5 | 4.5 | 1.6 ± 0.4 | 0.18 ± 0.05 | 10 | 0.016 ± 0.003 | 0.005 ± 0.001 | 0.03 | – | 0.017 | 0.006 | 0.06 | 5 | 1.2 |
| | B2 | 80–100 | 5.8 | 4.8 | 1.21 ± 0.28 | 0.12 ± 0.04 | 12 | 0.025 ± 0.005 | 0.005 ± 0.001 | 0.08 | – | 0.015 | 0.003 | 0.10 | 29 | 0.3 |
| | Oi | 0–1 | 4.7 | 3.6 | 444 ± 16 | 12.9 ± 1.4 | 40 | 5.57 ± 1.11 | 0.19 ± 0.04 | 10.23 | 1.542 | 2.314 | 0.051 | 14.14 | 22 | 63.3 |
| | Oe,pyr | 1–3 | 4.1 | 3.1 | 375 ± 13 | 11.7 ± 1.3 | 37 | 1.57 ± 0.31 | 0.07 ± 0.01 | 4.14 | 0.430 | 0.997 | 0.062 | 5.63 | 9 | 64.3 |
| | E$_{pyr}$ | 3–8 | 4.2 | 3.3 | 29 ± 4 | 0.95 ± 0.19 | 36 | 0.10 ± 0.02 | 0.004 ± 0.001 | 0.08 | 0.022 | 0.027 | 0.010 | 0.14 | 4 | 3.3 |
| 23AP | E | 8–14 | 4.7 | 3.9 | 5.3 ± 1.2 | 0.23 ± 0.07 | 27 | 0.05 ± 0.01 | – | 0.01 | 0.005 | 0.004 | – | 0.02 | 2 | 1.2 |
| | Bs | 14–30 | 5.4 | 4.8 | 9.7 ± 2.2 | 0.51 ± 0.10 | 22 | 0.04 ± 0.01 | – | 0.01 | 0.009 | 0.017 | 0.003 | 0.04 | 3 | 1.3 |
| | B1 | 30–75 | 5.5 | 5.2 | 2.3 ± 0.5 | 0.18 ± 0.05 | 15 | 0.025 ± 0.005 | – | 0.03 | 0.011 | 0.018 | 0.004 | 0.06 | 8 | 0.8 |
| | B2 | 75–120 | 6.1 | 5.0 | – | – | – | 0.019 ± 0.004 | – | 0.09 | 0.018 | 0.017 | 0.006 | 0.13 | 13 | 1.0 |
| | Oi | 0–3 | 4.6 | 3.6 | 439 ± 15 | 9.3 ± 1.0 | 55 | 5.75 ± 1.15 | 0.26 ± 0.05 | 6.28 | 1.352 | 2.487 | 0.194 | 10.32 | 27 | 38.6 |
| | Oe, pyr | 3–5 | 4.6 | 3.3 | 357 ± 12 | 9.2 ± 1.0 | 45 | 1.48 ± 0.30 | 0.07 ± 0.01 | 1.17 | 0.215 | 0.434 | 0.120 | 1.94 | 9 | 22.7 |
| | E$_{pyr}$ | 5–11 | 4.8 | 3.9 | 27 ± 4 | 0.78 ± 0.16 | 40 | 0.08 ± 0.02 | 0.003 ± 0.001 | 0.03 | – | 0.040 | 0.020 | 0.09 | 5 | 1.8 |
| 45AP | E | 11–20 | 5.6 | 4.2 | 2.5 ± 0.6 | 0.17 ± 0.05 | 17 | 0.03 ± 0.01 | 0.003 ± 0.001 | 0.03 | – | 0.023 | 0.021 | 0.07 | 31 | 0.2 |
| | Bs | 20–45 (50) | 5.7 | 4.8 | 5.2 ± 1.2 | 0.34 ± 0.10 | 18 | 0.020 ± 0.004 | – | 0.08 | – | 0.030 | 0.021 | 0.13 | 22 | 0.6 |
| | B1 | 45 (50)–70 | 6.0 | 4.9 | 2.3 ± 0.5 | 0.19 ± 0.06 | 14 | 0.015 ± 0.003 | – | 0.02 | – | 0.018 | 0.016 | 0.05 | – | – |
| | B2 | 70–90 | 5.8 | 4.0 | 1.8 ± 0.4 | 0.22 ± 0.06 | 10 | 0.04 ± 0.01 | 0.003 ± 0.001 | 3.53 | 2.080 | 0.219 | 0.097 | 5.92 | 54 | 10.9 |
| | B3 | 90–110 | 5.9 | 4.0 | 1.6 ± 0.4 | 0.16 ± 0.05 | 12 | 0.03 ± 0.01 | 0.004 ± 0.001 | 3.92 | 2.270 | 0.188 | 0.101 | 6.48 | 67 | 9.7 |
| | Oi | 0–1 | 4.8 | 3.6 | 395 ± 14 | 10.4 ± 1.1 | 44 | 4.64 ± 0.93 | 0.19 ± 0.04 | 9.75 | 1.128 | 1.883 | 0.033 | 12.80 | 34 | 38.2 |
| | Oe | 1–2 | 4.6 | 3.6 | 443 ± 16 | 10.1 ± 1.1 | 51 | 3.16 ± 0.63 | 0.17 ± 0.03 | 8.99 | 0.944 | 2.331 | 0.065 | 12.32 | 33 | 37.4 |
| | E$_{pyr}$ | 2–6 | 4.7 | 3.5 | 32 ± 5 | 1.01 ± 0.20 | 37 | 0.08 ± 0.02 | 0.004 ± 0.001 | 0.15 | 0.032 | 0.023 | 0.005 | 0.21 | 13 | 1.6 |
| 79AP | E | 6–16 | 5.0 | 3.9 | 6.2 ± 1.4 | 0.41 ± 0.08 | 18 | 0.038 ± 0.008 | 0.003 ± 0.001 | 0.02 | 0.013 | 0.009 | 0.002 | 0.04 | – | – |
| | Bs | 16–35 | 5.4 | 4.6 | 7.4 ± 1.7 | 0.45 ± 0.09 | 19 | 0.053 ± 0.011 | – | 0.04 | 0.027 | 0.021 | 0.016 | 0.10 | 9 | 1.1 |
| | B1 | 35–60 | 5.9 | 4.7 | 2.6 ± 0.6 | 0.19 ± 0.06 | 16 | 0.057 ± 0.011 | 0.004 ± 0.001 | 0.09 | 0.042 | 0.019 | 0.014 | 0.17 | 17 | 1.0 |
| | B2 | 60–90 | 5.9 | 4.7 | 1.4 ± 0.3 | 0.13 ± 0.04 | 13 | 0.020 ± 0.004 | – | 0.23 | 0.079 | 0.026 | 0.009 | 0.34 | 33 | 1.0 |
| | Oi | 0–2 | 4.4 | 3.2 | 460 ± 16 | 11.2 ± 1.2 | 48 | 2.28 ± 0.46 | 0.15 ± 0.03 | 6.93 | 1.091 | 2.045 | 0.054 | 10.12 | 24 | 41.8 |
| | Oe, pyr | 2–4 | 4.3 | 3.0 | 442 ± 16 | 8.9 ± 1.0 | 58 | 1.34 ± 0.27 | 0.09 ± 0.02 | 4.10 | 0.507 | 0.613 | 0.040 | 5.26 | 11 | 48.0 |
| | E$_{pyr}$ | 4–10 | 4.9 | 3.3 | 11.8 ± 2.7 | 0.42 ± 0.08 | 33 | 0.12 ± 0.02 | 0.006 ± 0.001 | 0.10 | 0.011 | 0.030 | 0.006 | 0.15 | 10 | 1.4 |
| 121AP | E | 10–20 | 5.1 | 3.8 | 3.1 ± 0.7 | 0.28 ± 0.08 | 13 | 0.040 ± 0.008 | 0.004 ± 0.001 | 0.03 | – | 0.005 | 0.001 | 0.04 | 17 | 0.2 |
| | Bs1 | 20–40 | 5.5 | 4.7 | 5.3 ± 1.2 | 0.32 ± 0.09 | 21 | 0.035 ± 0.007 | 0.004 ± 0.001 | 0.02 | 0.001 | 0.028 | 0.007 | 0.06 | 5 | 3.1 |
| | Bs2 | 40–60 | 5.6 | 4.6 | 2.6 ± 0.6 | 0.22 ± 0.06 | 19 | 0.022 ± 0.004 | 0.003 ± 0.001 | 0.04 | 0.007 | 0.027 | 0.005 | 0.08 | 7 | 1.3 |
| | B | 60–100 | 5.7 | 4.5 | <1.0 | <0.1 | 14 | 0.031 ± 0.006 | 0.007 ± 0.001 | 0.06 | 0.005 | 0.017 | 0.006 | 0.09 | 22 | 1.2 |

Note: $\sum$—is the sum of the exchange cations, BS—base saturation, CEC—cation exchange capacity, WSOC—water soluble organic carbon, WSON—water soluble organic nitrogen, dash—below the definition limit. B3 horizons contained, comparable to organic layers, levels of exchangeable cations (up to 6.5 mmol 100 g$^{-1}$ of the soil). Eluvial horizons (E and E$_{pyr}$) demonstrated the least sum of exchangeable cations.

The pH$_{H2O}$ values of soil mineral horizons ranged from 4.7 to 6.1. There is the common tendency of decreasing soil acidity with the soil depth in all soil profiles, as the pH values increase from 4.2–4.9 in the upper mineral horizons (E$_{pyr}$ and E) to 5.7–6.1 in the deepest horizons (Bs and B). A fire impact on soil acidity at upper mineral horizons (E$_{pyr}$ and E) is not recognized even one year after the fire (1AP site). Despite the low total projective cover (TPC) of ground vegetation, an early restoration stage of post-fire succession (23AP site) is characterized by a rather strong acidification of both E$_{pyr}$ and E horizons (pH 4.2 and 4.7, respectively).

The studied soils have similar soil texture and a weak profile differentiation of texture fraction contents. The upper soil horizon presented by sandy and loamy sands deposits are underlain by silty clay and clay loam deposits. The increasing clay content up to 19–24% from 70 cm was revealed only for the 45AP site. The upper coarse-textured horizons are characterized by a high sandy fraction and vary from 85 to 98%; the content of the clay fraction was 0–4%; low content of the silty fraction (1–6%) was detected. There was no distinct pyrogenic effect on the texture of upper mineral horizons.

The organic horizons are characterized by the highest sum of exchangeable cations, varying from 10 to 14 mmol 100 g$^{-1}$ of the soil in upper Oi horizons and from 1.9 to 12.2 in Oe,pyr (Table 2). The content of exchangeable cations decreased 10–100-fold in the mineral soil horizons (0.02–0.17 mmol 100 g$^{-1}$ of the soil), except the 45AP site where there was clayey B2.

Calcium cations generally predominate among the exchangeable bases (60 ± 15%, up to 82%). The highest variation of exchangeable Ca$^{2+}$ concentrations observed for the 1AP

site: from 0 in E, and Bs1 and Bs2 horizons to 10.3 mmol 100 g$^{-1}$ found in the pyrogenic Q$_{pyr}$ horizon. The exchangeable Mg$^{2+}$ content ranges among soils from 0.0 to 2.3 mmol 100 g$^{-1}$ of the soil (15 ± 10%). The exchangeable K$^+$ can be the only base detected in soils such as in the E and Bs1 horizons of the 1AP site and varies from 0.001 to 2.5 mmol 100 g$^{-1}$ (24 ± 17%). Sodium in general has the least content (7 ± 8%) among bases varying from 0 to 0.19 mmol 100 g$^{-1}$, and only for the 45AP site, its contribution to the sum bases reaches 32% in the Bs horizon. The base saturation is uneven throughout the soil profiles in all studied plots and varies from 0 to 67%.

Specific surface area (SSA) was very different among the soil horizons and studying soils (Figure 2b). The highest values were observed in organic (0.224–0.618 m$^2$ g$^{-1}$) and spodic illuvial Bs horizons (0.772–1.221 m$^2$ g$^{-1}$). Among the organic horizons, the Q$_{pyr}$ horizon at the site 1AP demonstrated the highest SSA (0.586 ± 0.021 m$^2$ g$^{-1}$). The SSA in the organic horizons of the rest sites (23–121 years after the fire) ranged from 0.224 ± 0.006 to 0.429 ± 0.008 m$^2$ g$^{-1}$ with a decrease in SSA with time elapsed after a fire. Among the mineral horizons, the lowest values were found in the eluvial E$_{pyr}$ (0.045–0.117 m$^2$ g$^{-1}$) and E horizons (0.031–0.221 m$^2$ g$^{-1}$).

The SSA has strong correlation values with the content of exchangeable Ca$^{2+}$ (r = 0.99, $p < 0.05$), Mg$^{2+}$ (r = 0.99, $p < 0.05$), K$^+$ (r = 0.98, $p < 0.05$), Na$^+$ (r = 0.96, $p < 0.05$), their sum (r = 0.99, $p < 0.05$) and CEC (r = 0.94, $p < 0.05$). In addition, the soil texture demonstrated a strong influence on SSA in mineral soil; the SSA has correlation with silt and clay contents (r = 0.98, $p < 0.05$).

### 3.4. Carbon and Nitrogen Content, Water Soluble Organic Matter

Carbon and nitrogen concentrations in the studied Podzols vary from 1.0 to 460 g kg$^{-1}$ and from 0.1 to 19.4 g kg$^{-1}$, respectively (Table 2). There is the common pattern in the distribution of the SOC (Figure 3a), C:N ratio (Figure 3b) and stable isotopes of C and N (Figure 3c,d) throughout the Podzol soil profile as shown for diverse sites 1 and 121.

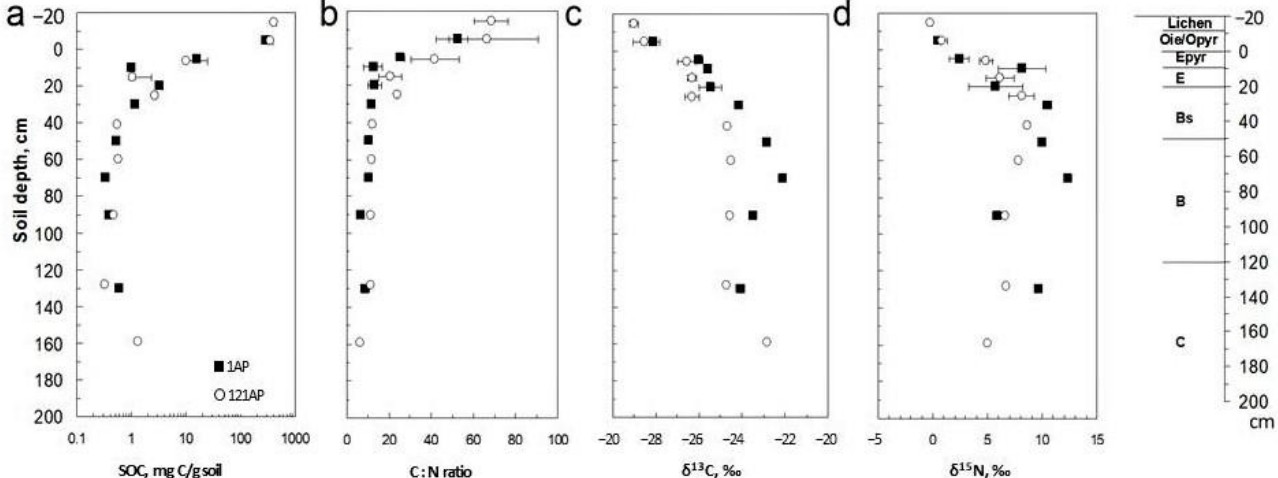

**Figure 3.** Changes of SOC content (**a**), C:N ratio (**b**), δ$^{13}$C-SOC (**c**) and δ$^{15}$N (**d**) with soil depth in pine forests affected by wildfires 1 and 121 years ago.

The largest concentrations of both C$_{tot.}$ and N$_{tot.}$ appear in soil organic horizons and they decrease 10-fold below in the upper mineral E$_{pyr}$ horizon. Nevertheless, there is an indication of increased carbon (11.8–34 g kg$^{-1}$) and nitrogen (0.4–1.1 g kg$^{-1}$) content in these pyrogenic horizons (E$_{pyr}$). Soils of all study plots demonstrated considerable depletion of eluvial E horizons with SOM (both C$_{tot.}$ and N$_{tot.}$) and its accumulation below in an illuvial spodic Bs horizon. In deeper subsoil horizons, the content of both C$_{tot.}$ and N$_{tot.}$ remain relatively constant at levels of 0.5 ± 0.1 and 0.05 ± 0.2 g kg$^{-1}$ of soil, respectively. Similarly, the C:N ratio decreases from 66 ± 24 in the organic layers to 20 ± 2

in the Bs horizon and further to $13 \pm 2$ in the deeper B horizons (Figure 3b). The C:N ratio decreased in the $Q_{pyr}$ horizon (1AP) after a fire ($p < 0.05$).

The upper soil horizons ($E_{pyr}$ and E) were divided into three fractions using a heavy liquid (fPOM $_{<1.6}$, oPOM $_{<1.6}$, MaOM $_{>1.6}$). MaOM $_{>1.6}$ fractions predominate in the studied horizons (Table 3). Its content varied from 91.91 to 98.45% in pyrogenic horizons $E_{pyr}$ and 98.53–99.76% in E horizons. The content of light fractions did not exceed 4%. Pyrogenic horizons $E_{pyr}$ contained from 0.73 to 5.44% of fractions fPOM $_{<1.6}$ and from 0.25 to 3.98% of fractions oPOM $_{<1.6}$. The content of the fPOM $_{<1.6}$ in the E horizons was 0.11–0.49%, and the oPOM $_{<1.6}$ varied from 0.06 to 0.73%. It is established that the pyrogenic horizons $E_{pyr}$ contained, in general, 2–29 times more fractions, fPOM $_{<1.6}$, and 1–66 times more fractions, oPOM $_{<1.6}$, than the horizon E. Despite the smaller amount, the main carbon pool is contained in fPOM $_{<1.6}$ and oPOM $_{<1.6}$ fractions. The carbon content in the fPOM $_{<1.6}$ varied from $44 \pm 7$ to $379 \pm 13$ g kg$^{-1}$; in the oPOM$_{<1.6}$ carbon content varied from $15 \pm 3$ to $512 \pm 18$. The content of nitrogen in the densimetric fractions is significantly lower than that of carbon. Fractions fPOM $< 1.6$ contained $1.6 \pm 0.3$ to $8.2 \pm 0.9$ g kg$^{-1}$. The oPOM$_{<1.6}$ fractions contained 0.91–10.5 g kg$^{-1}$. Fraction MaOM $_{>1.6}$ contained 1.0–6.70 g kg$^{-1}$ carbon and 0.1–0.26 g kg$^{-1}$ nitrogen.

**Table 3.** Contribution of the fraction; carbon and nitrogen content in densimetric fractions of $E_{pyr}$ and E horizons.

| Site, Horizons | | fPOM $_{<1.6}$ | | | | oPOM $_{<1.6}$ | | | | MaOM $_{>1.6}$ | | | |
| --- | --- | --- | --- | --- | --- | --- | --- | --- | --- | --- | --- | --- | --- |
| | | Mass, % | C | N | C/N | Mass, % | C | N | C/N | Mass, % | C | N | C/N |
| | | | g kg$^{-1}$ | | | | g kg$^{-1}$ | | | | g kg$^{-1}$ | | |
| 1AP | $E_{pyr}$ | 2.16 | $358 \pm 13$ | $7.5 \pm 0.8$ | 56 | 0.54 | $350 \pm 12$ | $9.7 \pm 1.1$ | 42 | 97.74 | $1.40 \pm 0.03$ | 0.1 | 16 |
| | E | 0.49 | $44 \pm 7$ | $1.6 \pm 0.3$ | 32 | 0.42 | $15 \pm 3$ | $0.91 \pm 0.18$ | 19 | 99.07 | 1.0 | 0.1 | 12 |
| 23AP | $E_{pyr}$ | 3.75 | $319 \pm 11$ | $6.4 \pm 0.7$ | 58 | 0.25 | $512 \pm 18$ | $10.2 \pm 1.1$ | 59 | 94.44 | 1.0 | 0.1 | 12 |
| | E | 0.35 | $293 \pm 29$ | $5.1 \pm 1.0$ | 67 | 0.08 | $208 \pm 21$ | $4 \pm 0.8$ | 61 | 98.82 | $1.250 \pm 0.029$ | 0.1 | 15 |
| 45AP | $E_{pyr}$ | 1.30 | $379 \pm 13$ | $5.8 \pm 1.2$ | 76 | 1.06 | $45 \pm 7$ | $1.8 \pm 0.4$ | 29 | 97.20 | $1.60 \pm 0.04$ | 0.1 | 19 |
| | E | 0.48 | $42 \pm 6$ | $1.23 \pm 0.25$ | 40 | 0.39 | $16 \pm 4$ | $1.38 \pm 0.28$ | 14 | 99.41 | 1.0 | 0.1 | 12 |
| 79AP | $E_{pyr}$ | 0.73 | $363 \pm 13$ | $8.3 \pm 0.9$ | 51 | 0.52 | $385 \pm 14$ | $9.8 \pm 1.1$ | 46 | 98.45 | $1.60 \pm 0.04$ | 0.1 | 19 |
| | E | 0.4 | $154 \pm 15$ | $4.1 \pm 0.8$ | 44 | 0.73 | $30 \pm 4$ | $1.7 \pm 0.3$ | 21 | 98.53 | 1.0 | 0.1 | 12 |
| 121AP | $E_{pyr}$ | 2.17 | $278 \pm 28$ | $5.6 \pm 1.1$ | 58 | 0.54 | $103 \pm 10$ | $4 \pm 0.8$ | 30 | 97.99 | 1.0 | 0.1 | 12 |
| | E | 0.45 | $154 \pm 15$ | $3.8 \pm 0.8$ | 47 | 0.08 | $29 \pm 4$ | $1.6 \pm 0.3$ | 21 | 99.75 | 1.0 | 0.1 | 12 |

The stable isotope ratios of carbon ($\delta^{13}C$) and nitrogen ($\delta^{15}N$) of bulk soil demonstrated the enrichment by heavy isotopes with soil depth (Figure 3c,d). In the most depleted organic layers of mature forests, $\delta^{13}C$ and $\delta^{15}N$ vary in the range from −27.9 to −29.9‰ and from −1.7 to 1.6‰, respectively.

The pyrogenic Q horizon of 1AP was characterized by the opposite response of $\delta^{13}C$ (enrichment) and $\delta^{15}N$ (depletion) to fire impact in comparison with intact organic layers. The soil horizons most enriched in heavy isotopes are the Bs and upper B horizons, where $\delta^{13}C$ ranges from −22.8 to −24.2 ‰ and $\delta^{15}N$ from 6 to 12 ‰. Deep mineral soil OM contains slightly more negative $\delta^{13}C$ and $\delta^{15}N$. There is an apparent relationship between C:N with $\delta^{13}C$ (Figure 4a) and $\delta^{15}N$ (Figure 4b) values, i.e., the decrease of the C:N ratio corresponds to an enrichment of SOM in heavy C and N isotopes. Organic layers composed of plant debris are characterized by the highest C:N ratios and lowest $\delta^{13}C$ values close to the initial C3 plant-derived matter and $\delta^{15}N$ of $NO_3$ deposited in northern latitudes with precipitation (1 to 3‰).

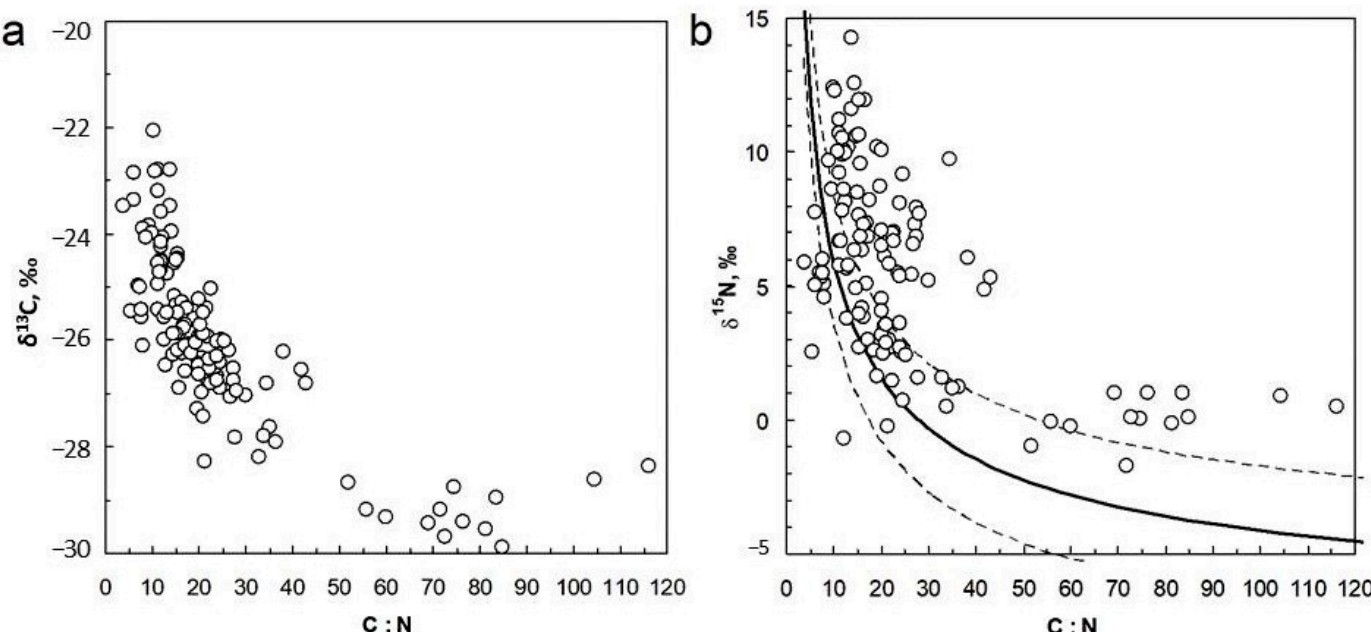

**Figure 4.** Relationships between C:N ratio and $\delta^{13}$C -SOC (**a**) and $\delta^{15}$N (**b**) in soils (O, E$_{pyr}$, E, Bs, B horizons) developed under pine forests of Central Siberia affected by wildfires 1–121 years ago. In the figure b, the bold continuous line is a function fitted to the unperturbed soil samples and dashed lines are the range ±2.4‰ as given in Connen et al. 2013 [48].

In mineral soil horizons narrowing C:N ratio with depth, mirroring the mineralization of SOM coincides with the predominant loss of light isotopes of both C and N. In regard to the relationship between C:N and $\delta^{15}$N, the values of the latter are shown to cover a relatively narrow value at any particular C:N ratio (bold and dashed lines on Figure 3b) in soils with little or no perturbation of the N cycle by the creation of new leaks or supplies of N [48]. Soil samples above the uncertainty envelope (set at ±2.4‰) indicate an accelerated N loss, whereas samples below indicate an accelerated N gain. Our findings demonstrated that the largest portion of $\delta^{15}$N values of Central Siberian Podzols lie above the uncertainty envelope (Figure 3d), demonstrating predominantly net losses of N as shown previously for frequently fire-affected soils of Yakutia (Eastern Siberia) [46].

The fire impact and long-term changes of SOM properties (C:N ratio) (Figure 5a), $\delta^{13}$C (Figure 5b) and $\delta^{15}$N (Figure 5c) in the upper 30 cm of soil (E$_{pyr}$, E and Bs horizons) during a stand restoration (1–121 year after a fire) was most evident in the eluvial horizons (E$_{pyr}$ and E). Spodic Bs horizons of studied soils did not demonstrate significant time-related differences in SOM properties.

The C:N ratios in the E$_{pyr}$ and E horizons of the recently fire-affected plot 1AP was 1.7-fold lower ($p < 0.05$) in comparison to older age plots (23AP and older) (Figure 5a). The SOC in E$_{pyr}$ and E horizons was enriched by heavy isotopes right after a fire (1AP) and $\delta^{13}$C tended to decrease in both horizons up to 23 years since a fire impact (23AP), then stabilizing at values $-26.3 \pm 0.3$‰ (45AP, 79AP and 121AP) (Figure 5b). SOM $\delta^{15}$N has appeared less affected by fire. Only E$_{pyr}$ horizons showed 1.5–3.0‰ lower $\delta^{15}$N values in the 1AP plot in comparison to older age plots. Recovery to pre-fire values occurred already in plot 23AP (Figure 5c), though the mid-age plot 45AP still demonstrated depletion in heavy isotopes.

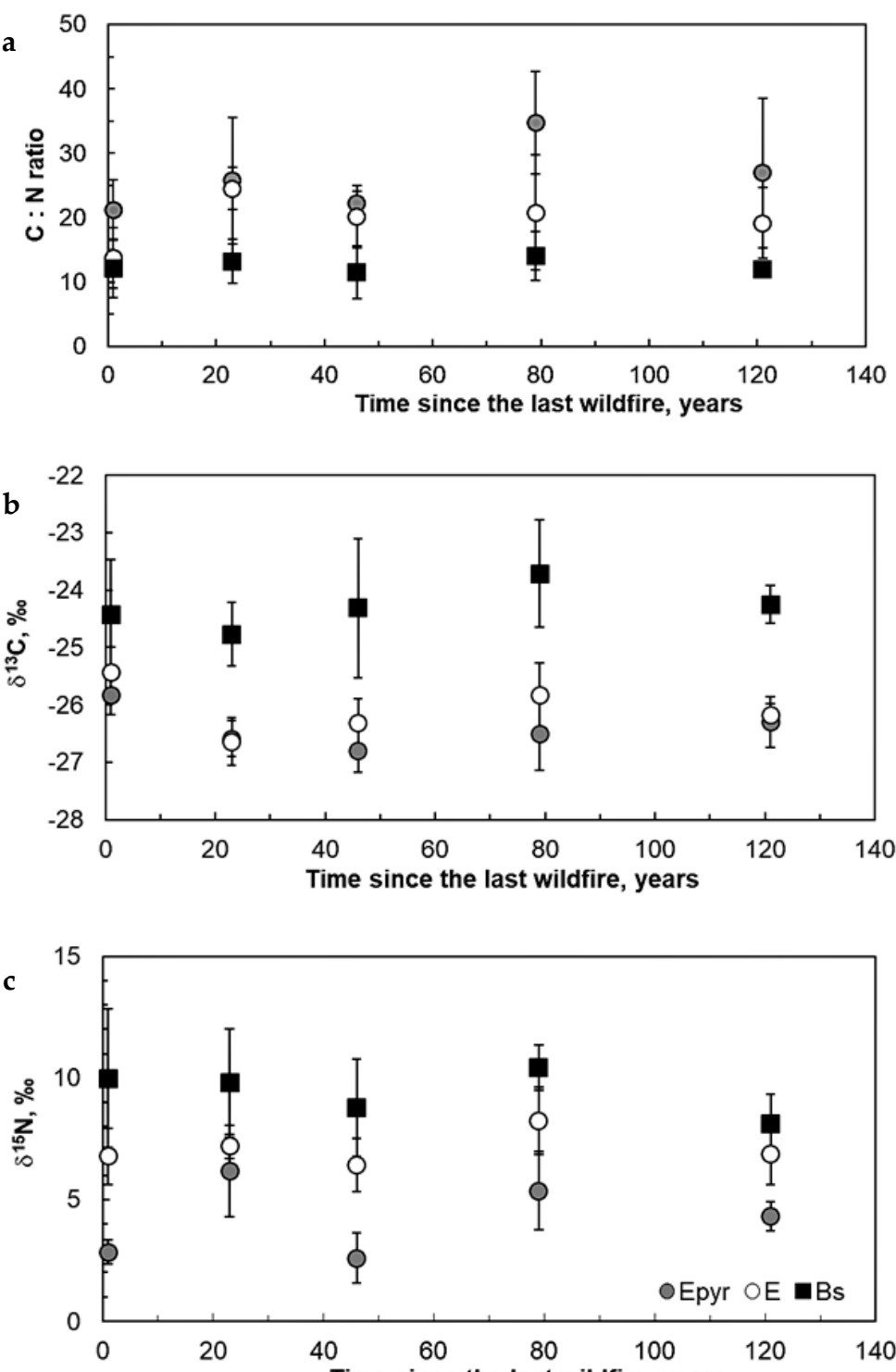

**Figure 5.** Post-fire dynamics of C:N ratio. (**a**), $\delta^{13}$C-SOC (**b**) and $\delta^{15}$N (**c**) in different soil genetic horizons: $E_{pyr}$, E (eluvial) and Bs (illuvial).

Organic horizons of the studied soils contain the largest concentrations of WSOC and WSON. Mineral horizons contain sufficiently less WSOC and WSON than organic horizons (Figure 6). It was found that the WSOC content in the area 1AP in the $Q_{pyr}$ horizon was only 0.5 mg g$^{-1}$, and the WSON content was 0.04 mg g$^{-1}$. The maximum WSOC content was revealed in the organic horizons of the soil 45AP. The content of WSOC in the organic horizons of plots 23AP and 45AP were 1.5–5.7 mg g$^{-1}$; the content of WSON varied from

0.07 to 0.26 mg g$^{-1}$. The content of water-soluble organic carbon in the E horizons of the soils of the studied areas was in the range of 0.08–0.10 mg g$^{-1}$. For sites 121 years after the last fire, lower contents of WSOC were found in the organic horizon (1.3–2.2 mg g$^{-1}$). However, in the E$_{pyr}$ of the soil of the plot 121 years after a fire, the maximum increase in the content of WSOC (0.12 mg g$^{-1}$) was revealed.

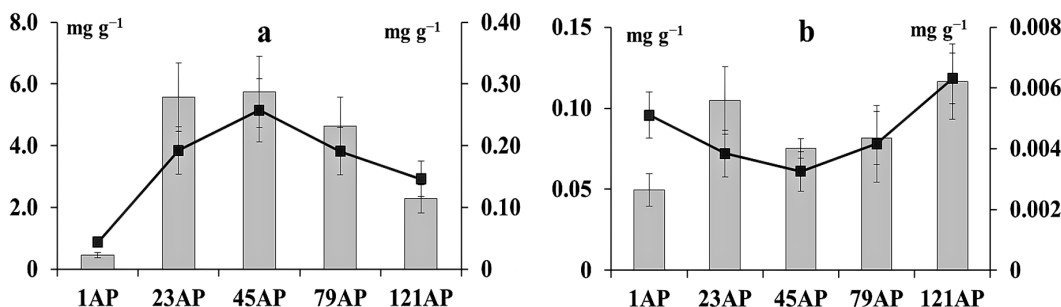

**Figure 6.** Water soluble organic matter at organic (**a**) and E$_{pyr}$ (**b**) horizons. Columns—water soluble organic carbon (WSOC) on the left scale; line—water soluble organic nitrogen (WSON) on the right scale.

*3.5. Content of PAHs of Soils*

PAHs were studied in the organic and topsoil mineral horizon (E$_{pyr}$) (Table 3) because PAHs were absent in the deeper mineral horizons [49].

We found a high content in all of the study horizons. The highest content was found for the 1AP soil. In the horizons of this soil, the PAH content reaches up to 3180 ng g$^{-1}$. There was a natural decrease in the mass fraction of PAHs with the time from the last fire event (Figure 7). An amount of 2–4 nuclear PAHs were prevalent in the organic horizons. The contribution of light structures to the total mass fraction of PAHs in organic horizons ranged from 80% in older burning sites to 95–96% in areas burned recently. The proportion of 2-nuclear PAHs decreased with age, with an increase in the age of the burning sites from 66 to 20%. The proportion of 5–6 nuclear PAHs increased.

The PAH content correlates with the total nitrogen content (r = 0.99, *p* < 0.05). A high correlation was found between total nitrogen with almost all PAHs (r = 0.51–0.99, *p* < 0.05). In addition, a reliable polynomial trend of decreasing PAHs over time after the last fire in the organic horizons (R$^2$ = 0.98) and E$_{pyr}$ (R$^2$ = 0.91) has been established (Figure 7a,b). Significant correlations were found between the time after the fire and the sum mass fraction of PAHs in the organic horizon (r= –0.71, p<0.05) and individual PAHs (r = −0.92 to r = 0.42, *p* < 0.05). For the pyrogenic mineral horizon E$_{pyr}$, we also found a negative relationship between age after a fire and the ∑PAH content in the is (r = −0.71, *p* < 0.05). With individual PAH compounds, the correlation coefficient varied: r = −0.69, . . . , 0,47, *p* < 0.05.

We calculated the series of PAH diagnostic ratios, which allowed us to determine the origin of the PAHs in soil: ANT/(ANT+PHE), PHE/ANT, (PYR+FLA)/(CHR+PHE), FLA/PYR, (PYR+BaP)/(PHE+CHR) and BaA/228 [50–54]. At the same time, these ratios were found to not be applicable for the studied soils. The ratios (PYR+BaP)/(PHE+CHR), (PYR+FLA)/(CHR+PHE) and BaA/228 showed a petrogenic origin of the PAHs in the burnt soils. The ratios PHE/ANT and ANT/(ANT+PHE) revealed a pyrogenic origin only for Qpyr(1AP) PAHs collected from the soil affected by fire 1 year ago. The ratio FL/PYR also inadequately reflected the origin of PAHs in this soil demonstrated petrogenic origin.

PAH ratios could demonstrate changes depending on the time from the last fire. For example, (PYR+BaP)/(PHE+CHR) was usually lower in the organic Q$_{pyr}$ (Oe, $_{pyr}$) horizons than in the E$_{pyr}$ horizons in old fire areas, while the opposite is typical for young burnt areas. The time patterns were also found for the ratio PHE/ANT, decreasing in the Q$_{pyr}$ horizons compared to E$_{pyr}$ in recent fire areas, and increasing in old fire areas.

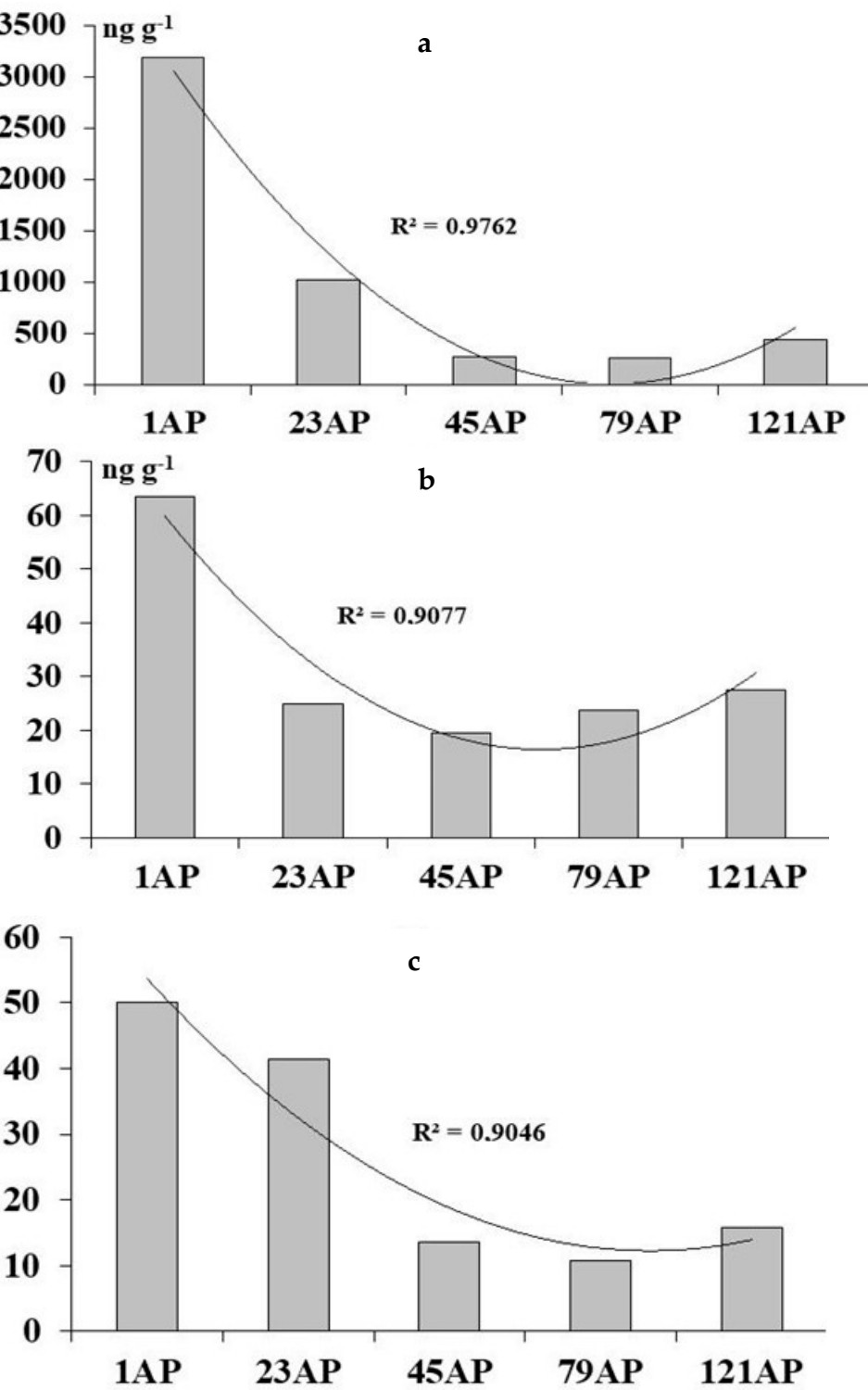

**Figure 7.** The total content of PAHs in the organic layer (**a**) and $E_{pyr}$ (**b**) of the studied soils. (**c**)—index of pyrogenic activity (IPA). The trend line is polynomial.

We have calculated a new pyrogenic activity index—IPA (Table 4, Figure 7c). The coefficient was the ratio $\sum$ PAHs at the organic horizon to $\sum$ PAHs at the upper mineral horizon. The IPA values gradually decrease in the time range after the fire from 50 at the 1 site to 11–16 at the 79 and 121 sites.

**Table 4.** PAH content at the upper soil horizons, ng g$^{-1}$.

| Site | Horizon | Depth, cm | 2-Ring | | 3-Ring | | | 4-Ring | | | | 5-Ring | | | | 6-Ring | | $\sum$PAHs | IPA * |
|---|---|---|---|---|---|---|---|---|---|---|---|---|---|---|---|---|---|---|---|
| | | | NP | ACE | FL | PHE | ANT | FLA | PYR | BaA | CHR | BbF | BkF | BaP | DahA | BghiP | IcdP | | |
| 1AP | Q$_{pyr}$ | 0–1 | 2007 | – | 97.2 | 424.3 | 59.9 | 36.8 | 58.8 | – | 305.6 | 64.4 | 11.8 | 48.2 | 58.8 | 7.6 | – | 3180.6 | 50.0 |
| | E$_{pyr}$ | 1–6 | 14 | – | 1.0 | 39.0 | 0.3 | 1.9 | 1.8 | – | 5.1 | – | – | – | – | – | – | 63.5 | |
| 23AP | Oe,$_{pyr}$ | 1–3 | 581 | – | 20.4 | 250.4 | 14.0 | 25.8 | 47.2 | – | 42.7 | 11.5 | 5.6 | 10.8 | 13.4 | 1.9 | – | 1024.6 | 41.3 |
| | E$_{pyr}$ | 3–8 | 4.3 | – | 2.0 | 12.1 | 0.7 | 3.1 | 2.0 | 0.2 | 0.3 | – | – | – | – | – | – | 24.8 | |
| 45AP | Oe,$_{pyr}$ | 3–5 | 127 | – | 7.8 | 75.6 | 3.9 | 17.1 | 5.0 | 2.0 | 9.2 | 7.3 | 2.0 | 5.3 | 3.4 | – | – | 266.1 | 13.6 |
| | E$_{pyr}$ | 5–11 | 6.5 | – | 1.3 | 6.7 | 0.4 | 2.6 | 0.8 | 0.3 | – | – | – | – | – | – | – | 19.6 | |
| 79AP | Oe,$_{pyr}$ | 1–2 | 75 | – | 7.0 | 121.3 | 5.2 | 9.5 | 5.4 | 1.5 | 2.5 | 6.1 | 3.2 | 3.6 | 14.7 | 1.8 | – | 256.6 | 10.8 |
| | E$_{pyr}$ | 2–6 | 8.9 | – | 1.2 | 6.8 | 0.3 | 3.5 | 2.1 | – | – | – | – | – | – | – | – | 23.8 | |
| 121AP | Oe,$_{pyr}$ | 2–4 | 191 | – | 9.0 | 94.5 | 3.8 | 8.8 | 4.0 | 0.8 | 22.8 | 62.0 | 0.9 | 5.8 | 29.3 | – | – | 432.7 | 15.7 |
| | E$_{pyr}$ | 4–10 | 13.2 | – | 0.7 | 7.5 | 1.0 | 1.7 | 0.7 | 0.4 | 2.2 | – | – | – | – | – | – | 27.5 | |

Note: 2–4 nuclear PAHs (NP—naphthalene, ACE—acenaphthene, FL—fluorine, PHE—phenanthrene, ANT—anthracene, FLA—fluoranthene, PYR—pyrene, BaA—benzo[a]anthracene, CHR—chrysene), 5–6 nuclear PAHs (BbF—benzo[b]fluoranthene, BkF—benzo[k]fluoranthene, BaP—benzo[a]pyrene, DahA—dibenzo[a,h]anthracene, BghiP—benzo[g,h,i]perylene, IcdP—indeno [1, 2, 3-c, d]pyrene), * IPA–ratio $\sum$ PAHs at organic horizon to $\sum$ PAHs at upper mineral horizon (E$_{pyr}$), dash—below the definition limit.

## 4. Discussion

### 4.1. Vegetation Dynamics, Morphological and Chemical Soil Properties

The species diversity and total projective cover of ground vegetation layers of plant communities depend on the age of the last fire. The cover of the herb-dwarf shrub layer was minimal immediately after the fire and further increased during post-pyrogenic succession. The restoration of the moss-lichen layer after-fire occurs in much faster rates than vascular plants. Recently, it was found that the middle-term post-fire recovery rate of the boreal moss-lichen layer is faster than the other layers of plant communities both in terms of species diversity [55] and biomass [56]. In this study, we also revealed faster moss-lichen layer restoration already at the 23-year site with it maintaining a stable state during further succession.

The investigated soils are characterized by a typical morphological profile (O–E–Bs–B) with the pyrogenic features in the upper genetic horizons (O$_{pyr}$, E$_{pyr}$), which is similar to the literature data on Podzols of different regions of the Russian Federation [9,57,58]. Under low intensity surface fires, changes in the vegetation and the burning of the organic horizon result in the formation of the pyrogenic Q$_{pyr}$ horizon. Similar pyrogenic soil properties have been noted in the soils of the Baikal region [59], which indicated the influence of surface fires on the transformation of organic soil horizons into new specific pyrogenic horizons.

The capacity and water content of organic soil horizons are significant indicators of forest soil restoration after fire impact. The moss-lichen layer of plants has not yet formed in the soils of young burnt areas. In these areas, the minimum values of humidity and thickness of organic horizons were revealed. The increasing of the thickness of organic horizons and the moisture content of organic soil horizons is found with the time that has passed since the last fire. The exception is the site 79AP, in which the moisture is lowest from the studying sites. In general, post-pyrogenic successions of lichen pine forests are characterized by a gradual increase in the proportion of mosses (primarily *Pleurozium schreberi*). The average thickness of forest litter at the stage of stabilization after fires for pine forests is 7.5–8.5 cm, which corresponds to the results published previously [60].

The degree of transformation of soil organic compounds is determined by the combustion temperature [24]. According to [25,61], natural biopolymers change already at temperatures of about 300 °C, and the formation of ash and coal begins. Coal particles play a significant role in the genetic horizons of soils for diagnosing past-fires [19,62]. Pyrogenic morphological features (charcoal, soot, partially charred residues) can be preserved both in organic, and in the upper mineral, soil horizons. The presence of pyrogenic signs is usually denoted by adding a «$_{pyr}$» to the designation of the name. In all the studied podzols, the presence of coal particles in the litter and eluvial horizons was revealed, which persisted for a long time after the fire, even after 121 years.

The Podzols of the studied areas are characterized by high acidity, typical for boreal landscape soils. A decrease in the acidity (increase at pH values) of organic horizons occurred in the first year after the fire, corroborating earlier observations in temperate and boreal forests [63,64]. Higher pH values are due to saturation by ash elements [21] and a reduction in the content of water soluble organic acids. In addition, coal particles are formed in which low-molecular-weight organic compounds can be absorbed [65,66]. In the course of succession, the vegetation of the ground cover is restored, and soil acidification is observed.

The most conservative properties of the studied soils, such as texture, are slightly subject to changes during low-intensity fires. Fire impacts on the SSA of organic horizons is most clearly seen in the first years after the fire [35]. The influx of coal particles with a high surface in the first years after a fire can have a significant impact on the chemical composition of soil water. With further succession and the supply of fresh plant material, the values decrease significantly.

### 4.2. Soil Organic Matter of Studied Soils

Bulk soil organic C and total N concentrations are among parameters that are regularly assessed in forest ecosystem research to analyze soil organic matter (SOM) pools and ecosystem development. Furthermore, stoichiometric ratios such as the carbon:nitrogen (C:N) ratio and stable isotope composition ($\delta^{13}C$ and $\delta^{15}N$) of SOM also provide a powerful tool for investigating spatial and temporal SOM dynamics and particularly, SOM turnover and stability, including fire disturbances [67–70].

Forest soils are characterized by the continuous inputs of fresh plant litter and roots that are steadily mixed and undergo microbial decomposition downward the soil profile [71–73]. Our data clearly demonstrate that Albic Podzols under pristine and fire-affected pine forests are supplied by a litter fall containing wide C:N and $^{13}C$ and $^{15}N$ depleted organic matter, as it was shown earlier for boreal and temperate forests [68,70]. With increasing soil depths and the aging of SOM [74], the content of C and N and C:N ratios in forest soils tend to decrease, and, in opposite, $\delta^{13}C$ and $\delta^{15}N$ show a trend toward enrichment by heavier isotopes [67–70]. These depth patterns are specific in Albic Podzols as the eluvial E horizon lying below an organic layer is strongly depleted by C and N in comparison to the deeper illuvial Bs horizon [15]. Despite this fact, $\delta^{13}C$ and $\delta^{15}N$ values in studied Podzols under pine forests demonstrate steady enrichment with soil depth, with relatively stable values in deep B and C horizons. It suggests the most intense microbial processing of plant residues in the upper 50 cm of soil (i.e., E and Bs horizons), resulting in the preferential release of 13C-depleted molecules [70,72]. The increase of $^{15}N$ in the soil depth gradient is the result of isotopic fractionation during SOM mineralization and the removal of depleted inorganic N via plants, microbes or leaching related to microbial driven processes and particularly, nitrification responsible for losses of $^{15}N$-depleted nitrate [75–79]. The greater contribution of microbial derived $\delta^{13}C$- and $\delta^{15}N$-enriched matter in deep soil [67,72,80] is another factor responsible for heavier isotopic content of subsoil SOM. On the other hand, stabilization or even decrease of $\delta^{13}C$ and $\delta^{15}N$ in deep SOM of Albic Podzols might reflect specifics of deep SOM turnover in this environment and requires further research. Thus, both $\delta^{13}C$ and $\delta^{15}N$ values are found mechanistically linked through the SOM mineralization and microbial processing, highlighting the importance of both parameters to determine the degree of organic matter turnover in soil [68,69,81–84].

Forest fires affect the total amount and cycling of C and N in an ecosystem, which are stored in the forest floor [85–87]. Common shifts in isotopic composition of remaining SOM, i.e., enrichment of topsoil with heavier isotopes [80], are demonstrated due to the predominant burning of the depleted top litter layer [86,88,89]. In regard to $^{15}N$ discrimination against heavier isotopes during volatilization of N [78] or the enhanced nitrification process and nitrate, leaching occurred after a fire impact and was shown in several boreal and temperate forests [48,70]. The wildfires in studied pine forests had relatively minor effects on the SOC content and isotopic composition of C and N in the remaining organic

layer after a fire, which is likely modulated by vegetation composition, fire intensity and severity [90]. On the other hand, the remaining OM on the mineral soil of studied pine forests demonstrated a narrower C:N ratio, which is usually significantly reduced after a wildfire, as a result of the combustion of upper fresher plant material's Oi layer [70,85,88].

Despite the changes in subsoil C and N content rarely being observed after a fire [70], the major consequence of periodic fires in pine forests is the formation of a specific $E_{pyr}$ horizon relatively enriched by C and having a wider C:N ratio, in comparison to typical E horizons of Podzols. These findings reflect the accumulation of charcoal, which in part might represent the ecosystem legacy carbon [91] and likely the most resistant pool of C in studied ecosystems which can contribute to C sequestration over longer (i.e., decadal to millennial) time scales [92]. The post-fire dynamics of the C:N ratio and isotopic composition of SOM in E and $E_{pyr}$ horizons demonstrated similar patterns of increasing C:N ratios and depletion of $^{13}C$ with time, reflecting an input of fresh OM. Changes in SOM were much less apparent in deeper soil, and the Bs horizon did not demonstrate any changes after the fire and further with post-fire succession.

Simultaneous measurements of $\delta^{15}N$, N and the C:N ratio in SOM are demonstrated to provide the indication for a perturbation to N cycling caused by a fire or anthropogenic influences [48,78,79]. Particularly, changes in N cycling are mostly related with the loss or gain of $^{15}N$-depleted substances which then result in larger or smaller $\delta^{15}N$ values of SOM at the observed C:N ratio [48]. An accelerated mineralization of soil organic matter may increase or decrease $\delta^{15}N$ in relation to respective C:N ratios, which in turn reflects recent or past soil perturbations. Generally, soil $\delta^{15}N$ appearing above the uncertainty value (set at $\pm 2.4‰$) supposes an accelerated N loss, and soil $\delta^{15}N$ below indicates an enhanced N gain. Analyzed Albic Podzol soils (Figure 4b) showed common patterns of past fire perturbations of SOM turnover; the majority of samples appeared above this uncertainty envelope, demonstrating the predominant net loss of N in studied pine forests.

It was found that the content of WSOC and WSON in post-pyrogenic organic horizons ($Q_{pyr}$) of «young» fire sites significantly decreases. The patterns of carbon content in water-soluble compounds after fires are determined by two main factors. The concentration of water-soluble carbon compounds is largely related to the excretions of ground cover plants. In the course of post-pyrogenic successions, this factor will be decisive. Fires lead to the death of ground cover plants, which in turn contributes to a significant decrease in the concentration of water-soluble carbon compounds [93]. As the plants of the top cover are restored, there is a gradual increase in the content of carbon and nitrogen of water-soluble organic compounds due to the input of organic substances [22]. The second factor is related with the coals formed during the fire. Coals have a significant surface area and can absorb a significant part of organic compounds entering the soil [94–96].

Densiometric fractionation of soil organic matter is devoted to the consistent simplification of the catfish system and the production of functional pools of organic matter with characteristic properties. Less destructive methods are used to isolate SOM fractions with specific properties [46,47,97,98]. The method makes it possible to divide the soil into fractions differing in composition, activity of participation in the biological carbon cycle and residence time in the soil into active "young" carbon, which is part of the light fractions of free ($fPOM_{<1.6}$) and occluded ($oPOM_{<1.6}$) organic matter, and slow "old" carbon of the heavy fraction ($MaOM_{>1.6}$) [99–101]. In the composition of the densimetric fractions of the studied upper mineral soil horizons, heavy fractions ($MaOM_{>1.6}$) predominate, which corresponds to the literature data [102–104]. The isolated densitometric fractions have some differences in the content of total organic carbon. Fractions $fPOM_{<1.6}$ and $oPOM_{<1.6}$ are characterized by higher carbon concentrations. The increase in carbon in the light fractions may be due to the enrichment of these horizons with pyrogenically modified parts of the litter. Pyrogenic carbon (PyC) is considered to be one of the most stable carbon deposition pools from the atmosphere [105]. We have suggested that light fractions can concentrate pyrogenic carbon better than heavy fractions. Some authors note in their works that pyrogenesis products are mainly part of light fractions [7,106].

It is known that fires change the chemical composition of organic matter, and lead to the enrichment of soils with pyrogenic black carbon (PyC) [107–109]. Our results on WSOC are close to the data of Ide with co-authors [58], and it can be assumed that some of the compounds formed during fires may have good solubility in water and affect the composition of WSOC in soils [110,111]. However, this requires additional studies of dynamic contents of the molecular composition of WSOC after fire impact.

### 4.3. Content of PAHs of Soils

Forest fires lead to the formation of large amounts of PAHs during combustion [36,112, 113]. We found similar patterns in PAH formation of organic horizons of Central Siberia and postpyrogenic boreal soils of the European North of Russia [12]. It should be noted that PAHs are quite good indicators of pyrogenic effects. The high content of PAHs in the organic horizon of the 1AP plot (one year after fire) supports the fact that these compounds have not yet been actively included in the biological cycle and are not decomposed in the process of photochemical destruction. This fact confirms that in the soils of the 1AP, PAHs are represented by two-nuclear structures by 66%, mainly naphthalene, which are most easily degraded. Other researchers [36,112,113] showed a high contribution of light PAHs to the $\sum$ PAHs in the organic horizons immediately after fire with a following strong decline during the year. Our results indicate that levels of light PAHs in the litter remain high 10–16 years after a fire [12]. The decomposition of naphthalene, the most vaporized and exposed to microbiological decomposition, results in the decline of the light PAHs fraction in soils with older fires. Microbiological decomposition of PAHs follows the pattern of successive rings' destruction. Light structures decompose first [67,114–116]. In the case of the site 121AP, an active decomposition of 3–4 nuclear PAHs to naphthalene has led to the increase of its concentration in the organic soil horizon. Heavy PAHs degraded to a lesser extent, and their proportion in the $\sum$PAHs was quite high (23%).

Factors affecting the pyrogenic formation of light and heavy PAHs are different [36,113]. The formation of 2–4 nuclear PAHs depends on combustion conditions, such as oxygen access, high temperature and exposure time. The formation of 5–6 nuclear structures mainly depends on the plant composition and type of burnt organic horizon. For instance, the organic horizon formed during *Pinus nigra* combustion contained less PAHs than those from *Pinus pinaster* [34]. The decrease in the mass fraction of PAHs in organic horizons was due to the effect of decomposition and restoration of vegetation, which leads to a change in the pyrogenic-modified $Q_{pyr}$ horizon. At the same time, the PAH content per unit mass at the organic horizons becomes less. Coefficients recommended for identifying the contribution of pyrogenic and pedogenic PAHs [50–54] were not correlated with the time of the last fire. This result may be due to the fact that these ratios of PAHs were calculated for the snow cover [50,117], and may not be completely suitable for soils. The increasing PAHs contents in soils could also be based on the processes of the anthropogenic burn of oil and gas. Thus, the natural biomass combustion PAHs could be included in the «petrogenic» PAHs.

The relationships between PAH ratios and fire age made it possible to calculate a new index—index of pyrogenic activity (IPA), most accurately reflecting the effect of the fire age on the composition of polyarenes in soils subjected to pyrogenic action. For forests affected by fires 1 and 23 years ago, the highest IPA ratio was revealed.

PAHs, especially heavy structures, are poorly water soluble [118,119]. Low migration of PAHs throughout the soil profile results in low PAH content in the mineral soil horizons under study. Minimal amounts of heavy PAHs were present in the mineral horizons. This result is in line with other studies [120], including those for podzols in the first months after the fire [49]. At the same time, there are few studies that demonstrate PAHs' migration into the underlying soil horizons [121].

## 5. Conclusions

Our study clearly shows that pyrogenic changes in the upper genetic horizons of Albic Podzols were saved over a long period of time from 1 year to 121 years after a fire. As a result of the study, it is shown that pine forests of the Eastern edge of western Siberia are exposed to fires, which affect the morphological and chemical properties of soils, and increase the contribution of pyrogenically modified components to the composition of the soil organic matter. The main morphological changes in soils after the fire are the formation specific soil horizons $Q_{pyr}$ and preservation pyrogenic sings at the Oe, $_{pyr}$ and $E_{pyr}$ horizons, which are saturated with pyrogenesis products (coals, soot). Pyrogenic inclusions are well preserved and diagnosed 121 years after the passage of the fire. The influence of forest fires in the early years on the chemical properties of Albic Podzols is manifested in a decrease in acidity, a decrease in the content of water-soluble forms of carbon and nitrogen and an increase in the content of light PAHs in organic and mineral horizons; there is also enrichment in heavier isotopes ($\delta^{13}$C). A new indicator is proposed—the pyrogenic activity index (IPA), which most accurately reflects the influence of the age of the last fire on the composition of polyarenes in soils exposed to pyrogenic effects.

**Supplementary Materials:** The following supporting information can be downloaded at: https://www.mdpi.com/article/10.3390/fire6020067/s1, Figure S1: Photos of landscapes and soil profiles of studied plots.

**Author Contributions:** Conceptualization, A.A.D. and A.S.P.; methodology, A.A.D.; software, A.V.P.; validation, V.V.S.; formal analysis, E.V.Y., D.A.S., E.Y.M., A.A.D. and V.V.S.; investigation, A.A.D., A.S.P. and V.V.S.; resources, A.A.D. and A.S.P.; data curation, A.A.D. and A.S.P.; writing—original draft preparation, A.A.D., V.V.S., Y.A.A.D. and A.S.P.; writing—review and editing, A.A.D. and A.S.P.; visualization, A.A.D. and A.S.P.; supervision, A.A.D.; project administration, A.A.D.; funding acquisition, A.A.D. All authors have read and agreed to the published version of the manuscript.

**Funding:** This work was supported by the Russian Foundation for Basic Research (RFBR) under Grant No. 19-29-05111 mk.

**Institutional Review Board Statement:** Not applicable.

**Informed Consent Statement:** Not applicable.

**Data Availability Statement:** The data presented in this study are available upon request from the corresponding author.

**Conflicts of Interest:** The authors declare no conflict of interest.

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
