# Peer review of "Fire-Induced Alterations of Soil Properties in Albic Podzols Developed under Pine Forests (Middle Taiga, Krasnoyarsky Kray)"

_fire, doi:10.3390/fire6020067_

Round 1

Reviewer 1 Report

The manuscript titled “Fire-induced alterations of soil properties in Albic Podzols developed under pine forests (middle taiga, Krasnoyarsky kray)” shows good results. The writing is very good. The results and discussion section reflect good acceptance. It needs only a minor modification before acceptance:

·        For a big audience, an appropriate flowchart of methodology is very much needed.

Author Response

Dear reviewer!

Thank you very much for your appreciation of the manuscript.

We have made some changes to the manuscript.

With best wishes, the authors of the article.

Reviewer 2 Report

This is a great work and provides important information of fire-affected changes in Albic Podzols at a long-term scale. 

Soil sampling, laboratory analyses, and data analysis are appropriate. Results are clearly presented. It looks this work good to be published.

Suggestions:

Fig1: add an overview map indicating the location of study area from world map.

Fig3: Add soil profile(s) which is helpful for readers to understand the soil properties. 

Author Response

Dear reviewer!

Thank you very much for your appreciation of the manuscript.

We have added photos of all soils at  supplementary information.

The location of the sites is shown in Figure 1, in addition, the article contains the coordinates of the sites.

With best wishes, the authors of the article.

Reviewer 3 Report

The manuscript entitled „Fire-induced alterations of soil properties in Albic Podzols developed under pine forests (middle taiga, Krasnoyarsky kray)“ submited by the group of Authors represents an interesting research on morphological and physicochemical properties and soil organic matter of Albic Podzols under pine forests of middle taiga zone of Siberia within a certain time frame. The forest fires had a significant influence on the soil properties and affected the morphological and chemical properties of soils, increased the contribution of pyrogenically modified components to the composition of soil organic matter.

The manuscript is well structured with clear aims. The Introduction was well elaborated. Some recent references can be included.

The methodology was well described and planned.

The results were extensively elaborated and well discussed. Authors should stress out the comment of pyrogenic black carbon on plants and soil microbial life.

The conclusion part gave the main conclusions. Conclusions were concentrated and well pointed.

Author Response

Dear reviewer!

Thank you very much for your appreciation of the manuscript.

We have added some additional references.

With best wishes, the authors of the article.